# OpenReview forum: "EvoScale: Evolutionary Test-Time Scaling for Software Engineering"
_ICLR.cc/2026/Conference — Submitted to ICLR 2026_

### Official Review · Reviewer_Gzv2 · 2025-10-30

**Soundness:** 2
**Presentation:** 2
**Contribution:** 2
**Rating:** 4
**Confidence:** 4

**Summary:**

This paper introduces EvoScale, a sample-efficient, test-time scaling method designed to resolve GitHub issues on SWE-Bench benchmark. The core idea of EvoScale is to iteratively refine outputs using an editor model. To ensure that each refinement step moves towards a better solution and without relying on verifiers at inference time, the authors employ RL, which trains the editor model to generate modifications that achieve a higher reward than the previous output. The authors claim that with EvoScale, a 32B parameter model can achieve performance comparable to that of models exceeding 100B parameters.

**Strengths:**

This paper focuses on using the key technique of test-time scaling to address the important and practical challenge of fixing GitHub issues.

**Weaknesses:**

1.The discussion of the problem and motivation is repetitive and redundant, taking up too much space and compressing the experiments section.

2. The proposed approach lacks novelty. The core techniques used, i.e., iteratively refining model outputs, are common, pre-existing methods.

3.The comparison in evaluation only includes naïve baselines like Reward Model Selection and Unit Tests Selection, lacking baselines from related work.

4.The technique may introduce additional overhead in terms of time and token consumption.

5.The paper lacks a thorough evaluation.

**Questions:**

1. The appendix provides a runtime comparison but only for a low sample budget. It would be more convincing to provide a table that shows both performance and runtime under the same sample sizes.

2.In Figure 4, why aren't the results for RL, Classical SFT, and Mutation SFT presented in a single, unified chart for easier comparison? The metrics (e.g., Best@40 vs. Best@30) and experimental settings (e.g., values of M and K) also seem to differ between the plots.

3.The main results for EvoScale-32B in Table 1 rely on both a reward model and unit tests, which potentially counteract the benefits of the RL-trained self-evolution. Could you report the performance of a pure EvoScale-32B without these external verifiers?

4.The results lack exact numbers to quantify the improvements and sample savings. It would be helpful to specify the exact performance gain and the number of samples saved by EvoScale-32B compared to a standard test-time scaling approach on the base model.

5.There are existing studies on reducing the cost of test-time scaling. It is suggested that you discuss and compare your method with these related works.

---

> ### Author Response · Authors · 2025-11-19
>
> Thank you for taking the time to review our submission. We address your concerns as follows,
>
> **1. The discussion of the problem and motivation takes too much space.**
>
> Thank you for the suggestion. **We have revised the manuscript accordingly**. Specifically, we moved the motivation discussion from the method section to the preliminaries section and reduced redundant explanations. In addition, we have included more experimental results in the main paper to improve completeness of analysis.
>
> **2. The core idea of iteratively refining output lacks novelty.**
>
> We acknowledges that the general idea of iterative refinement has appeared in prior work, such as [8]. However, **the main contribution of our paper is not only the conceptual framing, but also the training framework that enables an LLM to learn how to perform iterative refinement effectively**. To our best knowledge, the proposed potential-based reward shaping RL formulation is novel, and it gives the model the ability to self-evolve without relying on external verifiers at inference. We also provide a theoretical guarantee showing that our RL objective induces monotonic score improvement across iterations. **In general, our work identifies the right tool for addressing a key bottleneck in SWE: the high sample cost of test-time scaling.** We believe this insight itself is valuable to the research community, and has not been previously explored in this field.
>
> **3. Lacking baselines comparison rather than Reward Model Selection and Unit Tests Selection.**
>
> The main scope of our work is to propose a method that trains a SWE model to perform test-time scaling efficiently. We would like to clarify that in the SWE literature, very few prior works have explicitly studied test-time scaling, and among these, the only two widely adopted techniques are unit tests selection [1, 2, 3, 4, 6] and reward model selection [5, 6, 7], regardless of whether the scaffold is agentic or agentless. To the best of our knowledge, no other test-time scaling baselines exist in SWE beyond these two categories.
>
> **4. The technique may introduce additional overhead in terms of time and token consumption.**
>
> Any test-time scaling method inevitably incurs more cost than a single-pass greedy method. However, our goal is not to match greedy decoding cost, but to develop a test-time scaling approach that is significantly more efficient than existing test-time scaling strategies.
>
> We have demonstrated the runtime efficiency in the submission (also see rebuttal response-6). As for token consumption, EvoScale introduces minimal extra token overhead: only the input token length increases slightly due to appending the prior generated patches (see demo examples in Appendix B.).
>
>
> **5. The paper lacks a thorough evaluation.**
>
> Section 5.2 and Appendix A provides extensive empirical studies, including ablations on mutation SFT, analyses of RL-enabled self-evolution, monotonic reward improvement, direct comparisons against existing test-time scaling baselines, mutation temperature effects, comparisons against classical SFT test-time scaling, and runtime efficiency measurements.
>
> We would be happy to include additional experiments in the revision if the reviewer has specific suggestions or particular evaluation settings.
>
> **6. Question-1: runtime and performance comparison in a same table.**
>
> Thank you for the suggestion. In the revised version, **we have added a new figure that directly compares performance v.s. runtime budgets across different test-time scaling methods (see Figure 11 in the Appendix)**. As shown in the new figure, EvoScale consistently achieves higher performance while incurring much lower runtime overhead than existing test-time scaling approaches.
>
> **7. Question-2: In Figure 4, why aren't the results for RL, Classical SFT, and Mutation SFT presented in a single, unified chart for easier comparison?**
>
> These two figures analyze different questions and thus use distinct experimental settings.
>
> Figure 4a evaluates guided evolution of SFT models using a reward-model selector. The model generates 10 samples per iteration, and the reward model selects the top-5 to guide the next iteration (10 samples × 4 iterations). This experiment demonstrates that mutation SFT is essential for enabling guided evolutionary refinement, whereas classical SFT fails to improve across iterations even with a larger training set.
>
> Figure 4b evaluates pure self-evolution without any external selection (5 samples × 4 iterations). This mainly emphesizes the model’s intrinsic ability to refine its own outputs. The results show that only the RL-trained model can self-evolve, while mutation SFT alone cannot improve without reward-model guidance.

---

> ### Author Response · Authors · 2025-11-19
>
> **8. Question-3: In Table-1, report the performance of a pure EvoScale-32B without external verifiers.**
>
> Thank you for the suggestion. **We have added the results for pure self-evolution (i.e., without any reward-model or unit-test selectors) to Table 1**. Under this setting, EvoScale-32B achieves 38.6% using 5×4 = 20 samples, which remains more sample-efficient than several baseline methods, including Llama3-SWE-RL-70B.
>
> We also note that self-evolution and verifier-based selection are complementary mechanisms. Self-evolution increases the overall quality of generated patches, making the sample distribution concentrated around high-scoring regions. When combined with a verifier, these improved samples make the selection process to be more sample-efficient. Conversely, a verifier can further boost performance by selecting the optimal sample, albeit at introducing additional cost.
>
> **9. Question-4: Specify the exact performance gain and the number of samples saved by EvoScale-32B compared to a standard test-time scaling approach on the base model.**
>
> **We have included such comparison in Figure 8**. Specifically, we compare: (1) applying self-evolution using the RL-trained EvoScale-32B model, and (2) applying standard test-time scaling using an SFT-trained model based on the same backbone (Qwen2.5-Coder-32B-Instruct). We do not compare directly to the raw base model because, without training, it often fails to produce syntactically valid patch formats.
>
> In terms of performance and sample savings. The RL-trained EvoScale-32B surpasses the SFT model's Best@50 performance even at greedy decoding (N=1).Under equal sample budgets, EvoScale consistently achieves much higher accuracy, resolving over 40 more instances.
>
> **10. Question-5: Discuss and compare existing studies on reducing the cost of test-time scaling.**
>
> Thank you for the suggestion. **We have revised the manuscript to include a subsection in the Related Work section** discussing prior work on efficient test-time scaling. As we note in the revision, most existing studies focus on reasoning tasks and rely on frozen LLMs. In contrast, the main contribution of our work is to train the model itself to perform test-time scaling more efficiently.
>
> **Reference**
>
> [1] SWE-RL: Advancing LLM Reasoning via Reinforcement Learning on Open Software Evolution, Wei et al., NeurIPS 2025.
>
> [2] SWE-Fixer: Training Open-Source LLMs for Effective and Efficient GitHub Issue Resolution, Xie et al., ACL 2025.
>
> [3] CodeMonkeys: Scaling Test-Time Compute for Software Engineering, Ehrlich et al., 2025.
>
> [4] Agentless: Demystifying LLM-based Software Engineering Agents, Xia et al., 2024.
>
> [5] Lingma SWE-GPT: An Open Development-Process-Centric Language Model for Automated Software Improvement, Ma et al., 2024.
>
> [6] R2E-Gym: Procedural Environments and Hybrid Verifiers for Scaling Open-Weights SWE Agents, Jain et al., COLM 2025.
>
> [7] Training Software Engineering Agents and Verifiers with SWE-Gym, Pan et al., ICML 2025.
>
> [8] Scaling LLM Test-Time Compute Optimally can be More Effective than Scaling Model Parameters, Snell et al., ICLR 2025.
>
> ***
> If our clarifications address your concerns, we would greatly appreciate your consideration in updating the score to reflect what we believe are meaningful contributions to the community.

---

> ### Author Response · Authors · 2025-11-27
> **Following up on our rebuttal**
>
> Thank you again for your time and effort in reviewing our work. We wanted to follow up on your review and our rebuttal, and kindly ask if there are any remaining concerns you may have. If we could resolve your concerns, we hope you will consider updating your score accordingly. Thank you!

---

### Official Review · Reviewer_n7xD · 2025-10-31

**Soundness:** 3
**Presentation:** 4
**Contribution:** 2
**Rating:** 4
**Confidence:** 5

**Summary:**

The paper proposes **EvoScale**, an evolutionary test-time scaling framework for software engineering (SWE) tasks. It views multi-sample inference as an evolutionary process, where model outputs are iteratively refined through selection and mutation. To further reduce sampling cost, the authors train the model to self-evolve via reinforcement learning, internalizing the guidance of a reward model. Experiments on SWE-Bench-Verified show that a 32B model (Qwen2.5-Coder) can match or exceed the performance of 70B–100B models with far fewer samples. The presentation is clear and technically solid, and the empirical results are strong.

**Strengths:**

- Excellent execution and presentation. The paper is well written, with rigorous experimental setup, clean figures, and strong empirical evidence supporting the method’s effectiveness.
- Technically sound integration of evolutionary search and RL. The design of “mutation SFT + RL with potential shaping” is conceptually elegant and experimentally validated.
- Meaningful improvements in sample efficiency. EvoScale achieves notable gains under the same computational budget, demonstrating that smaller models can approach large-model performance with clever test-time strategies.
- Comprehensive analysis. The ablation studies, temperature experiments, and runtime comparisons are all thoughtfully conducted, providing convincing support for the claimed contributions.

**Weaknesses:**

- Directionally misaligned with the future of SWE research. The proposed approach remains strictly agentless—it treats issue resolution as a static, one-shot editing problem. While the method is effective within this paradigm, the field is rapidly shifting toward agentic and interactive systems (e.g., SWE-Agent) that perform multi-step reasoning, tool use, and environment feedback. Under this new paradigm, test-time scaling of static generation becomes a historical dead end rather than a forward-looking research direction.
- Limited real-world applicability. SWE tasks inherently involve dynamic context, runtime interaction, and multi-file dependencies. Test-time scaling assumes an independent scoring function (reward model or unit tests), which fails to capture these structured, process-dependent aspects. Thus, even with RL-based self-evolution, the framework remains detached from the actual software development workflow.
- Incremental conceptual novelty. While the “evolutionary” view is creative, it essentially reinterprets existing test-time compute scaling ideas (e.g., iterative refinement, verifier-guided search) rather than introducing a fundamentally new paradigm.
- Lack of adaptability beyond the static pipeline. The method is evaluated only in a pipeline-based setup with fixed retrieval, showing no evidence it can integrate into agentic or runtime environments—precisely where current research momentum lies.

**Questions:**

N/A

---

> ### Author Response · Authors · 2025-11-19
>
> Thank you for taking the time to review our submission. We thank the reviewer for appreciating our method's technical soundness and finding our empirical evluation comprehensive and convincing. Below we answer the rest of the questions:
>
> **1. Directional misalignment with agentic framework: Agentless is outdated.**
>
> First, **while agentic frameworks have gained significant attention, they are not universally superior for SWE tasks**. Prior agentic SWE works [1, 4] explicitly acknowledge some key limitations: (1) agentic systems often require long-horizon, multi-step trajectories, and (2) they are difficult to scale for smaller models. In contrast, agentless approaches tend to be more stable, less costly, and better suited for small-sized open-source models, which still remain widely used in some practical SWE settings.
>
> Second, **our proposed method is framework-agnostic. EvoScale can be integrated into either Agentless or Agentic workflows** because test-time scaling applies on the final patches, regardless of how they are produced (**See Rebuttal Response-4**).
>
>
> **2. Lack of real-world applicability: Test-time scaling with static scoring function is ineffective for real-world agentic workflows.**
>
> We respectfully disagree with this statement. Test-time scaling is orthogonal to the system scaffold, no matter the system is agentic or agentless. Although agentic workflows involve dynamic trajectories and runtime interactions, test-time scaling can always be applied at the trajectory level by generating multiple candidate trajectories and selecting the best final patch to submit. **In fact, two representative works [1, 2] have explicitly demonstrated the effectiveness of standard test-time scaling techniques (reward model selection) in Agentic SWE systems.**
>
>
> **3. Incremental novelty of "evolutionary" concept: similiar to exisiting iterative refinement methods.**
>
> We acknowledges that the general idea of iterative refinement has appeared in prior work, such as [3]. However, **the main contribution of our paper is not only the conceptual framing, but also the training framework that enables an LLM to learn how to perform iterative refinement effectively**. To our best knowledge, the proposed potential-based reward shaping RL formulation is novel, and it gives the model the ability to self-evolve without relying on external verifiers at inference. We also provide a theoretical guarantee showing that our RL objective induces monotonic score improvement across iterations. **In general, our work identifies the right tool for addressing a key bottleneck in SWE: the high sample cost of test-time scaling.** We believe this insight itself is valuable to the research community, and has not been previously explored in this field.
>
> **4. The proposed method cannot integrate into agentic SWE pipeline.**
>
> EvoScale can be easily integrated into Agentic workflows. To demonstrate this, **we have added new experiments in Appendix A.1 of the revised version, where we integrate EvoScale into the SWE-Agent scaffold**. Interestingly, even without any additional training, a strong model such as Seed-OSS-36B-Instruct already exhibits self-evolution capabilities and surpasses standard test-time scaling in both performance and sample efficiency (Figure 9.). Additionally, we observe that incorporating evolutionary test-time scaling allows the agent to use fewer interaction steps across evolution iterations (Figure 10.). We believe that applying our training framework (mutation SFT + RL) has the potential to further improve agentic systems, and we consider this a promising direction for future work.
>
> **Reference**
>
> [1] Training Software Engineering Agents and Verifiers with SWE-Gym, Pan et al., ICML 2025.
>
> [2] CodeMonkeys: Scaling Test-Time Compute for Software Engineering, Ehrlich et al., 2025.
>
> [3] Scaling LLM Test-Time Compute Optimally can be More Effective than Scaling Model Parameters, Snell et al., ICLR 2025.
>
> [4] SWE-smith: Scaling Data for Software Engineering Agents, Yang et al., 2025.
>
>
> ***
> If our clarifications address your concerns, we would greatly appreciate your consideration in updating the score to reflect what we believe are meaningful contributions to the community.

---

> > ### Comment · Reviewer_n7xD · 2025-11-26
> >
> > Thank you for the rebuttal and the additional explanations. I would like to clarify that my core concerns center on two fundamental issues that remain unresolved.
> >
> > First, the distinction between agentless and agentic SWE settings is not merely a matter of compatibility. Agentic workflows have become prominent because real SWE tasks typically involve long-horizon interaction, multi-step validation, and iterative correction. These characteristics are not naturally handled by agentless approaches. Indicating that EvoScale can operate within both frameworks does not fully address whether the method matches the structural requirements of modern SWE tasks.
> >
> > Second, although test-time scaling techniques can be applied to agentic trajectories, the key question is whether this reflects realistic usage. In practical SWE development, engineers do not generate many trajectories and choose the best one. They generally expect a system to produce an effective fix in a single attempt. Test-time scaling tends to improve Pass@K by increasing sample counts rather than enhancing the underlying capability of the model. Without evidence that the method can improve Pass@1 through training rather than sampling, its real-world impact remains uncertain.
> >
> > These two points form the basis of my remaining concerns, and the rebuttal does not significantly change my assessment.

---

> > > ### Author Response · Authors · 2025-11-26
> > > **Further Clarification**
> > >
> > > We thank the reviewer for further clarification. However, we respectively disagree with these two points.
> > >
> > > **Concern-1.**
> > >
> > > The reviewer acknowledges that EvoScale is compatible with both agentless and agentic settings, yet concludes that because agentic workflows are “more realistic,” any method that also works in an agentless setting is inherently misaligned with modern SWE practice. This reasoning does not follow. Compatibility with both settings strengthens, not weakens, the general applicability of the method.
> > >
> > > **Concern-2.**
> > >
> > > The argument that “engineers do not generate many trajectories and choose the best one” implies that research on test-time scaling is fundamentally inappropriate for SWE. This is inconsistent with existing literature. Prominent SWE test-time scaling methods explicitly rely on generating and selecting among multiple candidates: CodeMonkeys [1] samples multiple trajectories; SWE-Gym [2] performs reward-model–guided trajectory selection. Even in industry, as a powerful coding agent, AlphaEvolve relies on iterative multi-attempt refinement.
> > >
> > > Furthermore, the reviewer’s expectation that real users “generally expect a single-attempt fix” would rule out test-time scaling research across other domains. For instance, in math reasoning, it could be true that a student may ideally want the model to answer in one shot, yet substantial research work, such as majority vote, self-consistency, tree search, MCTS, investigates multi-sample (multi-step) test-time scaling in math reasoning. The same logic applies to SWE.
> > >
> > > Finally, whether test-time scaling is needed depends on task difficulty. For simple coding tasks (e.g., LeetCode) or middle-school math problems, one-shot solutions are indeed expected. For open-ended scientific discovery, hundreds of attempts are required. SWE lies between these extremes: many tasks are challenging enough that multi-sample test-time scaling could be beneficial, but efficiency is also important in practice. This is essentially the motivation behind EvoScale: to make test-time scaling efficient and practical rather than prohibitively expensive.

---

### Official Review · Reviewer_KMFj · 2025-10-31

**Soundness:** 3
**Presentation:** 3
**Contribution:** 3
**Rating:** 6
**Confidence:** 3

**Summary:**

This paper proposes EvoScale, an evolutionary test-time scaling method to improve small LLMs on real-world software engineering tasks. Instead of generating many candidate patches at once and selecting the best via a verifier, EvoScale treats code-editing as an iterative evolutionary process: generate candidates → select promising ones → condition on them → refine. The authors show that SFT models alone cannot perform mutation-style refinement, so they introduce a mutation-aware SFT phase, followed by potential-based RL training to enable models to self-evolve without external verifiers. They theoretically show monotonic improvement guarantees under a shaped reward. Experiments on SWE-Bench-Verified demonstrate that a 32B Satori-SWE model reaches 41.6%, matching or exceeding models >100B and much larger RL-trained systems while using significantly fewer samples. EvoScale provides efficient test-time scaling and shows sample efficiency benefits vs unit-test-driven or reward-model-driven selection. Overall, it advances agentless SWE pipelines by enabling self-refining LMs without runtime execution or expensive sampling.

**Strengths:**

1. Frames test-time scaling as evolution rather than brute sampling; aligns with natural mutation-refine workflows.
2. Potential-based reward shaping ensures monotonic improvement; it avoids sparse reward pitfalls.
3. 32B model reaches 41.6% on SWE-Bench Verified, beating or matching >100B models with far fewer samples
4. No expensive agent rollouts, minimal inference overhead, fewer unit-test calls.

**Weaknesses:**

1. More ablations isolating prompt effects would strengthen evidence for LM-as-mutation operator design.

2. Unit tests still needed for final selection in main benchmark. Despite "no verifier at inference" claims, final selection in SWE-Bench uses both RM + tests Clarify when EvoScale stands alone.

3. Iterative refinement can bias toward local optima; exploration vs exploitation trade-offs are not deeply analyzed.

**Questions:**

N/A

---

> ### Author Response · Authors · 2025-11-19
>
> Thank you for taking the time to review our submission. We thank the reviewer for appreciating our method's novelty and finding our method both effective and efficient. Below we answer the rest of the questions:
>
> **1. More ablations isolating prompt effects of LLM as mutation operator.**
>
> Thank you for the suggestion. In our experiments, we did not observe strong sensitivity of the LLM mutation operator to the prompt design itself. Our mutation prompt is a simple, hand-crafted template without any prompt optimization. Instead, we find that mutation SFT training is the key enabling effective iterative refinement. As shown in Figure 4 (a), models without mutation SFT are unable to improve over iterations, whereas mutation-SFT–trained models consistently exhibit improvement. This indicates that the mutation capability arises mainly from the training procedure, rather than prompt engineering.
>
>
> **2. Unit tests still needed for final selection in main benchmark.**
>
> We have added the results for pure self-evolution (i.e., without any reward-model or unit-test selectors) to Table 1. Under this setting, EvoScale-32B achieves 38.6% using 5×4 = 20 samples, which remains more sample-efficient than several baseline methods, including Llama3-SWE-RL-70B.
>
> We also note that self-evolution and verifier-based selection are complementary mechanisms. Self-evolution increases the overall quality of generated patches, making the sample distribution concentrated around high-scoring regions. When combined with a verifier, these improved samples make the selection process to be more sample-efficient. Conversely, a verifier can further boost performance by selecting the optimal sample, albeit at introducing additional cost.
>
> **3. Iterative refinement can bias toward local optima.**
>
> We agree that iterative refinement methods may risk converging to local optima. However, our experiments has investigated the methods that could mitigate this issue. In Figure 7, we conduct an ablation study on mutation sampling temperature and observe that using a higher temperature (e.g., 1.2) achieves the best performance. To further understand this phonomenon, Figure 13 analyzes the diversity of generated patches across refinement iterations. The results show that higher temperatures help preserve sample diversity, whereas low temperatures lead to near-zero diversity in later iterations.
>
> The key takeaway is that maintaining patch diversity is essential, and controlling sampling temperature is a simple yet effective mechanism. Other strategies, such as sampling more samples and filtering out redundant ones, could also be effective, albeit at introducing additional computational cost.

---

### Official Review · Reviewer_kYJN · 2025-11-01

**Soundness:** 3
**Presentation:** 3
**Contribution:** 2
**Rating:** 4
**Confidence:** 4

**Summary:**

This paper introduces EvoScale, a novel framework for software engineering (SWE) tasks that enhances model performance at test-time. It creatively reframes multi-sample inference as an evolutionary process, iteratively improving model outputs through selection and mutation. To make this process highly efficient, the authors employ reinforcement learning to train the model to "self-evolve," thereby internalizing the guidance of a reward model and reducing sampling costs. The framework's effectiveness is validated by strong empirical results on SWE-Bench-Verified, where a 32B model (Qwen2.5-Coder) achieves performance comparable or superior to 70B-100B models with significantly fewer samples. Overall, the work is presented clearly, is technically solid, and offers compelling findings.

**Strengths:**

1. Significant Gains in Sample Efficiency: The paper's primary contribution is a meaningful improvement in computational efficiency. EvoScale powerfully demonstrates that with clever test-time strategies, smaller models can achieve performance comparable to, or even exceeding, much larger models under the same computational budget.

2. Elegant Technical Integration: This efficiency gain is rooted in a technically sound integration of evolutionary search and Reinforcement Learning. The specific design, which combines "mutation SFT" with "RL with potential shaping," is both conceptually elegant and experimentally validated.

3. Rigorous and Comprehensive Analysis: The claims are convincingly supported by a series of thoughtful analyses. The comprehensive ablation studies, temperature experiments, and runtime comparisons robustly validate the design choices and the method's effectiveness.

4. Exceptional Clarity and Execution: Overall, the paper is exceptionally well-written and presented. The rigorous experimental setup, clean figures, and strong empirical evidence make the work both easy to follow and highly persuasive.

**Weaknesses:**

1. Directional Misalignment with the Future of SWE: The paper's primary weakness is its strategic focus on an outdated, agentless paradigm that treats software engineering as a static, one-shot editing problem. This direction is fundamentally misaligned with the field's rapid shift towards agentic, interactive systems that use multi-step reasoning and tool interaction. As such, improving static generation feels like a historical dead end rather than a forward-looking contribution.

2. Lack of Generalizability and Real-World Applicability: This agentless design inherently lacks the generalizability required for practical application. It is ineffective in real-world agentic workflows because it cannot handle dynamic context, runtime feedback, or multi-file dependencies. The reliance on a static scoring function makes the framework detached from the actual, process-dependent nature of software development.

3. Weak Empirical Support and Missing Baselines: The framework's limitations are reflected in its empirical results. The claimed improvements are not benchmarked against crucial SOTA models like Qwen3-Coder-30B or agentic systems like SWE-Smith. Furthermore, the fact that EvoScale-32B (Greedy) scores 35.8—nearly identical to its baseline (agentless + Qwen2.5-Coder-32B-Instruct)—casts serious doubt on the practical value of the proposed evolutionary mechanism.

4. Incremental Novelty Rather Than True Innovation: Ultimately, the "evolutionary" concept appears to be more of a re-interpretation than a true innovation. It reframes existing test-time compute scaling ideas (iterative refinement, verifier-guided search) without introducing a fundamentally new paradigm capable of addressing the challenges of modern agentic systems.

**Questions:**

N/A

---

> ### Author Response · Authors · 2025-11-19
>
> Thank you for taking the time to review our submission. We thank the reviewer for appreciating our method's technical soundness and finding our empirical evluation comprehensive and convincing. Below we answer the rest of the questions:
>
> **1. Directional misalignment with agentic framework: Agentless is outdated.**
>
> First, **while agentic frameworks have gained significant attention, they are not universally superior for SWE tasks**. Prior agentic SWE works [1, 4] explicitly acknowledge some key limitations: (1) agentic systems often require long-horizon, multi-step trajectories, and (2) they are difficult to scale for smaller models. In contrast, agentless approaches tend to be more stable, less costly, and better suited for small-sized open-source models, which still remain widely used in some practical SWE settings.
>
> Second, **our proposed method is framework-agnostic. EvoScale can be integrated into either Agentless or Agentic workflows** because test-time scaling applies on the final patches, regardless of how they are produced. To demonstrate this, we have added new experiments in Appendix A.1 of the revised version, where we integrate EvoScale into the SWE-Agent scaffold. Interestingly, even without any additional training, a strong model such as Seed-OSS-36B-Instruct already exhibits self-evolution capabilities and surpasses standard test-time scaling in both performance and sample efficiency (Figure 9.). Additionally, we observe that incorporating evolutionary test-time scaling allows the agent to use fewer interaction steps across evolution iterations (Figure 10.). We believe that applying our training framework (mutation SFT + RL) has the potential to further improve agentic systems, and we consider this a promising direction for future work.
>
> **2. Lack of real-world applicability: Test-time scaling with static scoring function is ineffective for real-world agentic workflows.**
>
> We respectfully disagree with this statement. Test-time scaling is orthogonal to the system scaffold, no matter the system is agentic or agentless. Although agentic workflows involve dynamic trajectories and runtime interactions, test-time scaling can always be applied at the trajectory level by generating multiple candidate trajectories and selecting the best final patch to submit. **In fact, two representative works [1, 2] have explicitly demonstrated the effectiveness of standard test-time scaling techniques (reward model selection) in Agentic SWE systems.**
>
>
> **3. Weak empirical support and missing SOTA baseline: Qwen3-Coder, SWE-Smith.**
>
> **We respectfully note that the reviewer has acknowledged the strong empirical performance of our method**, stating that "The framework's effectiveness is validated by strong empirical results on SWE-Bench-Verified." and "Empirical evidence make the work both easy to follow and highly persuasive." We appreciate this recognition and clarify the concerns raised about SOTA baselines.
>
> Qwen3-Coder-30B is a recently released model, and our contribution is orthogonal to specific model architectures. EvoScale introduces a training strategy (mutation SFT + RL) that can be applied to any open-source code model, including Qwen3-Coder.
>
> Regarding SWE-Smith, its main contribution is the construction of a large-scale open-source dataset for training SWE Agent through SFT. It does not propose a test-time scaling technique and therefore is not directly comparable to our method. Nevertheless, we would like to point out that SWE-Agent-LM-32B (built upon SWE-Smith data) reports 40.2% on SWE-Bench-Verified, while EvoScale-32B achieves comparable performance using fewer training samples.
>
> **4. EvoScale-32B (Greedy) scores 35.8, nearly identical to agentless + Qwen2.5-Coder-32B-Instruct.**
>
> **We respectfully point out that this statement is not accurate. To the best of our knowledge, there is no reported evidence showing that agentless + Qwen2.5-Coder-32B-Instruct can achieve 35.8% on SWE-Bench-Verified**. In our experiments, we observed that the base model Qwen2.5-Coder-32B-Instruct even fails to generate syntactically valid patches without SFT training. Even after standard SFT training on full training data, the resulting model fails to exceed 30% accuracy on SWE-Bench-Verified (see Figure 8), which is significantly below EvoScale-32B’s greedy score of 35.8%.

---

> ### Author Response · Authors · 2025-11-19
>
> **5. Incremental novelty of "evolutionary" concept: similiar to exisiting iterative refinement methods.**
>
> We acknowledges that the general idea of iterative refinement has appeared in prior work, such as [3]. However, **the main contribution of our paper is not only the conceptual framing, but also the training framework that enables an LLM to learn how to perform iterative refinement effectively**. To our best knowledge, the proposed potential-based reward shaping RL formulation is novel, and it gives the model the ability to self-evolve without relying on external verifiers at inference. We also provide a theoretical guarantee showing that our RL objective induces monotonic score improvement across iterations. **In general, our work identifies the right tool for addressing a key bottleneck in SWE: the high sample cost of test-time scaling.** We believe this insight itself is valuable to the research community, and has not been previously explored in this field.
>
>
> **Reference**
>
> [1] Training Software Engineering Agents and Verifiers with SWE-Gym, Pan et al., ICML 2025.
>
> [2] CodeMonkeys: Scaling Test-Time Compute for Software Engineering, Ehrlich et al., 2025.
>
> [3] Scaling LLM Test-Time Compute Optimally can be More Effective than Scaling Model Parameters, Snell et al., ICLR 2025.
>
> [4] SWE-smith: Scaling Data for Software Engineering Agents, Yang et al., 2025.
>
> ***
> If our clarifications address your concerns, we would greatly appreciate your consideration in updating the score to reflect what we believe are meaningful contributions to the community.

---

> ### Author Response · Authors · 2025-11-27
> **Following up on our rebuttal**
>
> Thank you again for your time and effort in reviewing our work. We wanted to follow up on your review and our rebuttal, and kindly ask if there are any remaining concerns you may have. If we could resolve your concerns, we hope you will consider updating your score accordingly. Thank you!

---

### Meta-Review · Area_Chair_vttP · 2026-01-05

**Summary:**

The paper introduces EvoScale, an evolutionary test-time scaling framework for software engineering tasks on SWE-Bench-Verified. By reframing multi-sample inference as iterative selection and mutation, and training models via mutation-aware SFT followed by potential-shaped RL to self-evolve without external verifiers, EvoScale enables a 32B model to achieve performance comparable to or exceeding much larger (>100B) models with far fewer samples.

Reviewers praised the technical elegance (integration of evolutionary search with RL, monotonic improvement guarantees), strong empirical results (significant sample efficiency gains), comprehensive ablations, and clear presentation. However, concerns focused on limited novelty (reframing existing iterative refinement/test-time scaling ideas), directional misalignment with the field's shift toward agentic/interactive systems, reliance on static scoring (unit tests/reward models) limiting real-world applicability, and insufficient evidence of generalization beyond agentless pipelines.

The authors' rebuttal and revisions addressed many points: added agentic integration experiments (SWE-Agent scaffold), pure self-evolution results, runtime comparisons, temperature/diversity analyses, and clarified complementarity of self-evolution with verifiers. These strengthened transparency but did not fully resolve debates on novelty or alignment with agentic trends.

**Reviewer Concerns:**

Technical details (ablations, runtime, integration) were largely addressed via revisions and new experiments. However, concerns on conceptual novelty, practical relevance in agentic/real-world workflows, and whether gains reflect deeper capability versus refined sampling remain partially outstanding.

**Reviewer Scores:**

Initial scores leaned borderline (two at 4, one at 6). Rebuttal yielded constructive dialogue but no confirmed upward adjustments, with consensus remaining cautious due to strategic direction questions.

---

### Decision · Program_Chairs · 2026-01-26

Reject